# The Ethanol Oxidation Reaction Performance of Carbon-Supported PtRuRh Nanorods

**Tzu-Hsi Huang [1], Dinesh Bhalothia [2], Shuan Lin [1], Yu-Rewi Huang [1] and Kuan-Wen Wang [1,\*]**

[1] Institute of Materials Science and Engineering, National Central University, Taoyuan 320, Taiwan;
candy10416@hotmail.com.tw (T.-H.H.); linshiuan@gapp.nthu.edu.tw (S.L.);
k03310401@hotmail.com (Y.-R.H.)

[2] Department of Engineering and System Science, National Tsing Hua University, Hsinchu 30013, Taiwan;
dinesh@mx.nthu.edu.tw

[\*] Correspondence: kuanwen.wang@gmail.com

**Abstract:** In this study, carbon-supported Pt-based catalysts, including PtRu, PtRh, and PtRuRh nanorods (NRs), were prepared by the formic acid reduction method for ethanol oxidation reaction (EOR) application. The aspect ratio of all experimental NRs is 4.6. The X-ray photoelectron spectroscopy and $H_2$-temperature-programmed reduction results confirm that the ternary PtRuRh has oxygen-containing species (OCS), including $PtO_x$, $RuO_x$ and $RhO_x$, on its surface and shows high EOR current density at 0.6 V. The corresponding physical structure results indicate that the surface OCS can enhance the adsorption of ethanol through bi-functional mechanism and thereby promote the EOR activity. On the other hand, the chronoamperometry (CA) results imply that the ternary PtRuRh has the highest mass activity, specific activity, and stability among all catalysts. The aforementioned pieces of evidence reveal that the presence of OCS facilitates the oxidation of adsorbed intermediates, such as CO or $CH_x$, which prevents the Pt active sites from poisoning and thus simultaneously improves the current density and durability of PtRuRh NRs in EOR.

**Keywords:** PtRuRh; ethanol oxidation reaction; nanorods; bi-functional mechanism; oxygen containing species

## 1. Introduction

Ethanol is one of the most promising sustainable energy in the future owing to its high energy density, non-toxicity, renewability, [1] and low green-house gas emission while produced [2,3]. However, the high bond dissociation energy of carbon–carbon bond (C–C) in ethanol is a critical challenge for ethanol oxidation reaction (EOR), which also hinders the commercialization of direct ethanol fuel cell (DEFC). Therefore, much effort has been made to promote the EOR performance of catalysts.

The oxidation of ethanol produces several products in which acetaldehyde and acetic acid are formed through 2 and 4 electrons ($e^-$) transfer, respectively [4]. However, the desire pathway is to form $CO_2$ as a product, which not only requires 12 $e^-$ transfer, but also involves the cleavage of the C-C bond. Several undesired intermediates, such as $CO_{ad}$ and $CH_{x,ad}$, may be produced during the 12 $e^-$ transfer, poisoning the state-of-the-art anode Pt catalysts [5–9]. Generally, EOR on Pt catalysts starts from the dissociative adsorption of ethanol as described in Equation (1).

$$CH_3CH_2OH + Pt \rightarrow Pt\text{-}CH_3CH_2OH_{ads} \tag{1}$$

Pt then will be poisoned by undesired intermediates as shown in Equation (2).

$$Pt\text{-}CH_3CH_2OH_{ads} \rightarrow Pt\text{-}CO_{ads} + CH_3 + 3H^+ + 3e^- \tag{2}$$

Therefore, in order to prevent the poisoning of Pt catalysts, binary catalysts are employed to improve the CO tolerance of Pt catalysts for EOR, especially alloying or modifying Pt with oxophilic metals, such as Sn [10–14] and Ru [10,14,15]. The presence of these oxophilic metal atoms can combine with oxygen-containing species (OCS), such as hydroxyl species ($OH^-$), which can further facilitate not only the incomplete oxidation of ethanol into acetic acid or $CO_2$ through so called bifunctional mechanism but also the anti-poisoning ability of Pt catalysts [16,17] described in Equations (3)–(5):

$$\text{Ru (or Sn)} + H_2O \rightarrow \text{Ru (or Sn)}-OH_{ads} + H^+ + e^- \tag{3}$$

$$\text{Pt-}CH_3CHO_{ads} + OH_{ads} \rightarrow \text{Pt} + CH_3COOH + H^+ + e^- \tag{4}$$

$$\text{Pt-}CO_{ads} + \text{Ru (or Sn)-}OH_{ads} \rightarrow \text{Pt} + \text{Ru (or Sn)} + CO_2 + H^+ + e^- \tag{5}$$

Therefore, alloying with oxophilic metal(s) is an efficient way to enhance the tolerance to poisoning intermediates for Pt catalysts. It has been reported that for [14] Pt/C, Pt-Ru (1:1) and Pt-Sn/C (3:1) catalysts, the oxidation of the adsorbed OCS is facilitated at lower potentials through supplying oxygen atoms from Ru or Sn oxides at an adjacent site.

The PtRu/C catalyst is the state-of-the-art anode catalyst in direct methanol fuel cells and DEFC, owing to its remarkable CO tolerance that is usually ascribed to the bifunctional mechanism and ligand effects. The bifunctional mechanism is ascribed to the facilitation of $CO_{ads}$ oxidation on Pt by OCS from Ru at low potentials, and the ligand effect suggests that the electronic structure of Pt is modified through formation of PtRu alloy, weakening CO adsorption on Pt [18–21]. However, Ru is easily dissolved into acid medium during the long-term operation of DEFC, which may degrade the CO tolerance and durability [22].

Besides PtRu/C catalysts, other binary and ternary catalysts, such as PtAu [23,24], PtAg [25,26], PtNi [27,28], PtRh [29,30], PtSnRh [31,32] and PtRuSn [33–35], have shown enhanced EOR performance attributed to the bifunctional mechanism and synergistic effect. Among them, PtRh/C [36,37] catalysts are regarded as the promising catalysts to decompose ethanol. Although theoretical studies suggest that Rh is not as active as Ru for OCS formation, Rh can enhance the EOR performance of Pt-based catalysts by synergistic effects and electron modification [38].

Moreover, the PtRuRh/C catalysts with different Pt/Ru/Rh compositions have been prepared for methanol oxidation reaction (MOR) [39,40]. It can be observed that the ternary PtRuRh can provide higher MOR activity than binary PtRu at low potential. Furthermore, the catalysts prepared at the ratio of $Pt_1Ru_1Rh_2$ possess the highest current density at 0.5 V (vs. RHE) in 0.5 M $H_2SO_4$ and 1 M $CH_3OH$ at 60 °C, suggesting the synergistic effect of Ru and Rh addition on their MOR performance. Therefore, ternary PtRuRh/C can be a promising EOR catalyst.

Moreover, controlling the morphologies of Pt nanostructure is a promising method to promote their catalytic activity, especially the structure-sensitive reactions, such as EOR [41,42]. The Pt 1-D structures, such as nanorods (NRs) and nanowires (NWs), exhibit higher catalytic activities toward MOR and EOR when compared with 0-dimensional NPs [43]. This can be attributed to its anisotropic nature, high aspect ratios, few lattice boundaries and low surface defect sites [44–46]. Pt NWs can display lower onset potential and higher current density than Pt NPs, attributed to a decrease in the activation energy for EOR of 1-D structures [47]. Similar enhanced EOR performances can be observed on PtRh NWs [48]. PtRh NWs have obviously lower onset potential and higher activity than PtRh NPs and Pt/C. After 2000 cycles, the PtRh NWs maintains 86% current density, which is the most durable catalyst among the prepared ones. It seems that 1-D structures such as NWs or NRs with second metals can possess higher activity and stability than NPs, which may be associated with unique structure and surface properties.

In order to prepare highly effective catalysts, in this study, carbon-supported PtRu, PtRh, and PtRuRh catalysts have been prepared by taking the advantages of the bifunctional mechanism and 1-D

structure. The presence of OCS can help the desorption of adsorbed intermediates on Pt active sites, which enhances the activity and stability of Pt catalyst, and 1-D structures with high aspect ratios can have better EOR performance than NPs, owing to its anisotropy and unique structure. The compositions, structures, morphologies, surface compositions and electrochemical performances of the prepared catalysts are analyzed by a field emission scanning electron microscope and an X-ray energy dispersive spectrometer (SEM-EDS), X-ray diffraction (XRD), transmission electron microscopy (TEM), X-ray photoelectron spectroscopy (XPS)/$H_2$-temperature-programmed reduction (TPR)/CO-stripping/cyclic voltammograms (CV), and linear sweep voltammograms (LSV)/ chronoamperometry (CA), respectively.

## 2. Experimental Section

### 2.1. Preparation of Carbon-Supported Pt and Pt-Based NRs

Carbon-support Pt, PtRu, PtRh, PtRuRh NRs with metal loading of 50 wt. % were prepared through the formic acid method (FAM). For of Pt NRs, 0.05 g carbon black (Vulcan XC-72R) was dispersed in the deionized the preparation (DI) water for half an hour and then aqueous solution of $H_2PtCl_6$ and formic acid were added into the above solution at 288 K for 3 days. The as-deposited solution was then filtered, washed with DI, dried in 333 K oven overnight, and named as Pt.

For the preparations of binary PtRu and PtRh, 0.05 g carbon black was dispersed in the DI, and then $H_2PtCl_6$ and formic acid were added into the solution at 288 K for 3 days. The second metal precursor ($RuCl_2$ or $RhCl_3$) and isopropanol (IPA) as a reductant were added into the as-deposited Pt solutions at 288 K for another 2 days. The as-deposited solution containing $RuCl_2$ or $RhCl_3$ was then filtered, washed with DI, dried in 333 K oven overnight, and named as PtRu and PtRh, respectively.

For the preparation of ternary PtRuRh catalysts, $RhCl_3$ was added into the as-deposited PtRu solution and reacted for another 2 days at 288K. The as-deposited solutions were filtered, washed with DI and dried at 333 K.

### 2.2. Characterizations of the Catalysts

The atomic compositions of catalysts were characterized by a field emission scanning electron microscope (JSM7000F) and an X-ray energy dispersive spectrometer (SEM-EDS, Bruker) operated at 15 kV. The phases and structures of prepared catalysts were analyzed by the X-ray diffraction (XRD) with CuK$\alpha$ radiation operated 40 kV and 25 mA at a scan rate of $3°$ min$^{-1}$. In order to calculate d-spacing, Bragg's law was applied:

$$n\lambda = 2dsin\theta \tag{6}$$

where n is positive integer, $\lambda$ is the wavelength of radiation source, $\theta$ is the diffraction angle and d is the d-spacing. In addition, for the face-centered cubic crystal style, the lattice parameter can be determined by the relationship of Equation (7):

$$d = \frac{a}{\sqrt{l^2+m^2+n^2}} \tag{7}$$

where l, m and n are the Miller index, d is the d-spacing value and a is the lattice parameter. In order to obtain the grain size, Schreer's equation was applied [49]:

$$\tau = \frac{K\lambda}{\beta COS\theta} \tag{8}$$

where the $\tau$ is mean grain size, K is constant, $\lambda$ is the wavelength of radiative source, $\beta$ is the full width at half maximum of the base peak and$\theta$ is the diffraction angle.

The morphologies of catalysts were confirmed by high-resolution transmission electron microscopy (HRTEM, JOEL-2100) operated at 160 kV. The catalysts powders were prepared by ultrasonically suspending in IPA. Afterward, the suspension was then dropped on 200 mesh copper grids.

The surface chemical states of various catalysts were analyzed by X-ray photoelectron spectroscopy (XPS, Thermo VG Scientific Sigma Probe) using an AlKα radiation at a voltage of 20 kV and a current of 30 mA. The surface compositions and chemical states of the samples are calculated by integral of each peak and a combination of Lorentzian and Gaussian lines was applied to fit the experimental curves. C 1s peak at 284.6 eV was used as an internal standard to determine the accurate binding energies.

The surface species of the prepared catalysts were characterized by the $H_2$-temperature-programmed reduction (TPR). The variations of hydrogen contents during the flowing of the reduction gas are detected by a thermal conductivity detector (TCD). The sample of 0.1 g was reduced by a flow of 20% $H_2$ in $N_2$ at the flow rate of 50 mL min$^{-1}$ while raising the temperature from 150 to 600 K at the heating rate of 7 K min$^{-1}$.

Electrochemical measurements through a CH Instruments Model 611c device were operated to analyze the performance of prepared catalysts. The counter and reference electrodes were Pt plate and saturated calomel electrode (SCE), respectively. The catalysts in IPA and Nafion solution (5 wt. %, DuPont) were dispersed and dropped on a glass carbon electrode as working electron. CO-stripping was operated by purging CO gas in 0.5 M $H_2SO_4$ for 30 min before the experiment; while operating, CO was kept purging at −0.1 V (vs. SCE) for 30 min, then CO-stripping was measured between −0.3 and 0.76 V (vs. SCE) at the rate of 50 mV s$^{-1}$ in $N_2$ saturated 0.5 M $H_2SO_4$ The electrochemically active surface area calculated form CO-stripping ($ECSA_{CO}$) is defined by the Equation (9):

$$ECSA_{CO} = \frac{Q_{CO}}{[Pt] \times 0.42} \tag{9}$$

where [Pt] presents the Pt loading on the electrode, $Q_{CO}$ indicates the charge for CO-desorption and 0.42 is assuming that the oxidation of a CO monolayer requires 0.42 mC/m$^2$ [50].

Cyclic voltammograms (CV) were swept between −0.24 and 1.0 V (vs. SCE) at the rate of 50 mVs$^{-1}$ in 0.5 M $H_2SO_4$ purged with $N_2$ for 30 min to ensure the electrolyte is $N_2$-saturated. The ECSA by H-adsorption ($ECSA_H$) [50,51] was calculated by integrating the areas of hydrogen desorption at 0–0.4 V ($Q_H$). The values were obtained from the following Equation (10):

$$ECSA_H = \frac{Q_H}{[Pt] \times 0.21} \tag{10}$$

where 0.21 (mC/m$^2$) is the charge required to oxidize a monolayer of $H_2$ on Pt active sites. For the EOR activity, the CVs were swept between −0.24 to 1.0 V (vs. SCE) with a scanning rate of 2 mVs$^{-1}$ in 0.5 M $H_2SO_4$ and 1 M $C_2H_5OH$ saturated with $N_2$. The durability tests of the catalysts were measured at consist voltage of 0.36 V (vs. SCE) for 2 hrs by chronoamperometric (CA), and the electrolyte for LSV and CA was both $N_2$-saturated 0.5 M $H_2SO_4$ containing 1.0 M $C_2H_5OH$.

## 3. Results and Discussion

The exact compositions of PtRu, PtRh, and PtRuRh determined by SEM-EDS are displayed in Table S1. It seems that by the FAM process, carbon supported PtRu, PtRh, and PtRuRh catalysts can be prepared.

The structural information of PtRu, PtRh, and PtRuRh catalysts obtained by XRD is shown in Figure S1. The peaks of XC-72R carbon black are located at 25° for all catalysts. For the Pt catalysts, peaks located at 39.80°, 46.28° and 67.53° are attributed to the (111), (200) and (220) planes of face-centered cubic (fcc) Pt, respectively, (JCPDS 870646). For PtRu and PtRh, the characteristic peaks are located at 40.23°, 46.70° and 68.03°, which are obviously shifted to larger angle when compared to the Pt reference, owing to the smaller atomic radius of Ru (134 pm) and Rh (134 pm) than that of Pt (139 pm) and lattice shrinkage during alloying. In terms of the ternary catalysts, the peaks are located at 40.63°, 47.07° and 68.40° for PtRuRh. The d-spacing calculation results are around 0.227, 0.224, 0.224, and 0.222 nm for Pt, PtRu, PtRh, and PtRuRh, respectively.

The decrease in d-spacing in binary and ternary catalysts demonstrates that Ru and Rh atoms partially substitute Pt atoms in the structure. The calculated lattice parameters for catalysts are listed in Table S1, in which the prepared binary and ternary catalysts have the lattice parameter of 0.384 nm compared to 0.392 nm for Pt, related to the alloying and EOR enhancement [52]. The calculated mean grain sizes for Pt, PtRu, PtRh, and PtRuRh are 5.3, 4.6, 4.6, and 4.8 nm, respectively.

Figure 1 presents the morphologies of Pt, PtRu, PtRh, and PtRuRh by HRTEM. From Figure 1a–d, it can be observed that various NRs can be successfully prepared by FAM, in which the mean aspect ratios of Pt, PtRu, PtRh, and PtRuR is 4.5 ± 0.9, 4.7 ± 0.9, 4.5 ± 0.8, and 4.7 ± 0.8, respectively. Figure 1e–h shows the morphologies after CA test, in which the mean aspect ratios of catalysts shorten into 2.1 ± 0.7, 2.1 ± 0.6, 1.8 ± 0.6, and 2.0 ± 0.7 for Pt, PtRu, PtRh, and PtRuRh, respectively, as summarized in Figure 1i–l, suggesting that during CA test, the catalysts suffer from dissolution, migration, corrosion, etc.

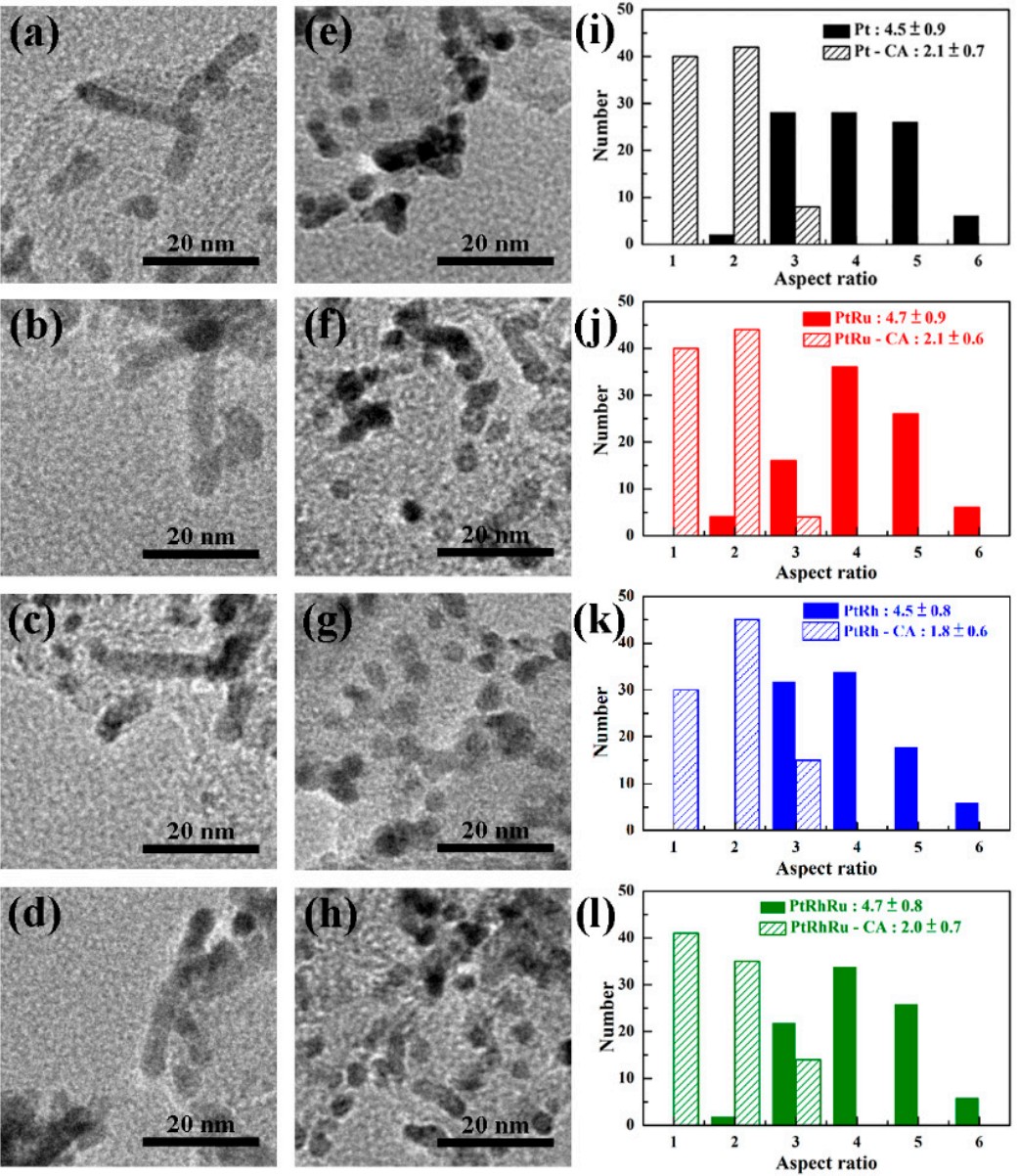

**Figure 1.** High-resolution transmission electron microscopy (HRTEM) micrographs of morphologies before CA: (**a**) Pt (**b**) PtRu, (**c**) PtRh and (**d**) PtRuRh; after CA: (**e**) Pt, (**f**) PtRu, (**g**) PtRh and (**h**) PtRuRh; aspect ratios distributions: (**i**) Pt, (**j**) PtRu, (**k**) PtRh and (**l**) PtRuRh.

The XPS spectra of Pt, PtRu, PtRh, and PtRuRh catalysts analyzed by XPS are displayed in Figure 2 and their surface compositions are listed in Table S2. In Figure 2, the Pt peaks located at 71.0 and 74.3 eV belong to $4f_{7/2}$ and $4f_{5/2}$, respectively. For other binary and ternary catalysts, slight shifts are noted due to different degrees of electron transfer between Pt and the second or third metals [53,54]. Moreover, the addition of the second metals influences the chemical states of Pt in which the Pt oxide compositions increase or decrease owing to the Ru or Rh addition, respectively. Meanwhile, the amount of OCS, such as $PtO_x$, $RuO_x$, or $RhO_x$, is also influenced and the EOR performances are also affected. It has been reported that the presence of Ru can modify the surface electronic structure of Pt, causing the downshift of the Pt d-band center, weakening the interaction between Pt and absorbed intermediate species, and enhancing the EOR durability of Pt-based catalysts [55]. Therefore, such OCS plays an important role in determining the EOR performance.

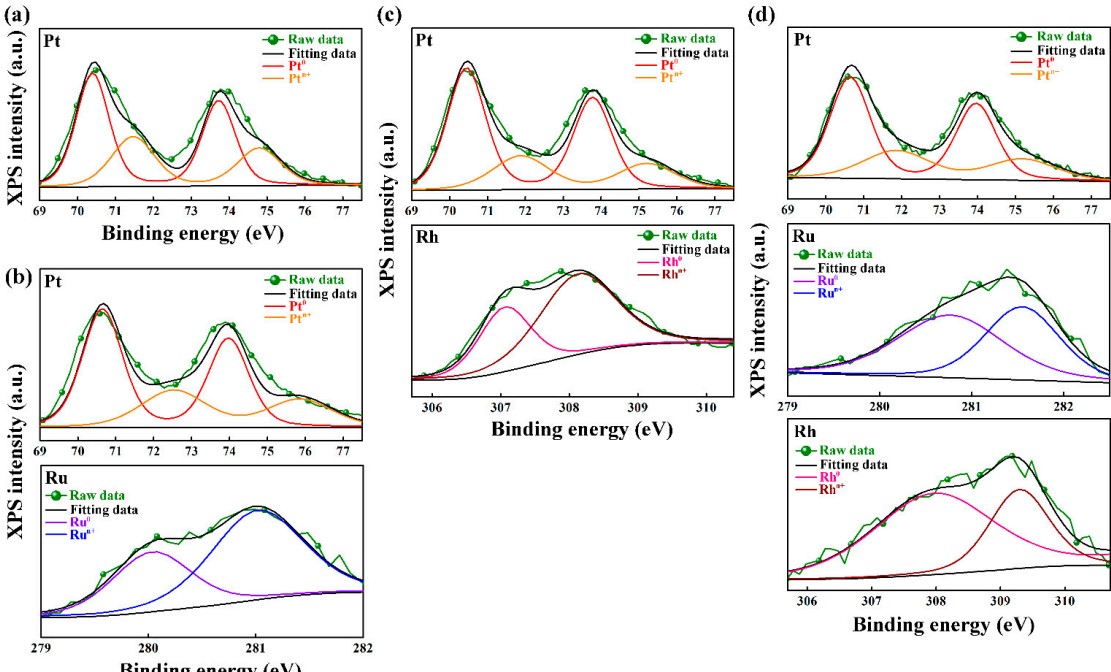

**Figure 2.** X-ray photoelectron spectroscopy (XPS) spectra of (**a**) Pt, (**b**) PtRu, (**c**) PtRh, and (**d**) PtRuRh.

On the other side, the TPR experiments are carried out to analyze the topmost surface species of the catalysts. The TPR spectra of PtRu, PtRh, and PtRuRh are displayed in Figure 3. The main reduction peaks for the catalysts located between 200 and 230 K are assigned to the reduction of the surface platinum oxides ($PtO_x$). For the reduction of ruthenium oxides ($RuO_x$) and rhodium oxides ($RhO_x$), the reduction peaks are located between 300 and 400 K [56,57] and above 500 K, respectively. The peak located at 250 K for PtRh may be owing to the reduction of surface PtRh alloy oxide ($A^{PtRh}$). Therefore, the main surface species of PtRuRh is Pt/Ru/Rh.

The CO-stripping experiments are performed, and the results for Pt, PtRu, PtRh, and PtRuRh are displayed in Figure 4a and Table S3. It is obvious that the onset potential of Pt is much higher than those of the binary and ternary catalysts, implying a more effective CO oxidation reaction for the latter ones. Overall, PtRu performs the lowest onset potential, followed by PtRuRh << PtRh [58–62], indicating that the incorporation of second or third metal can indeed enhance the CO oxidation by the bi-functional mechanism. On the other side, the $ECSA_{CO}$ is compared in Table S3 in which PtRuRh has the largest value among all catalysts, suggesting that PtRuRh can provide the largest Pt surface area for CO oxidation. Besides, it has been reported that $RuO_x$ helps the desorption of CO adsorbed on Pt [63]. Thus, it appears that PtRuRh is superior to promote the electro-oxidation of adsorbed CO than other samples.

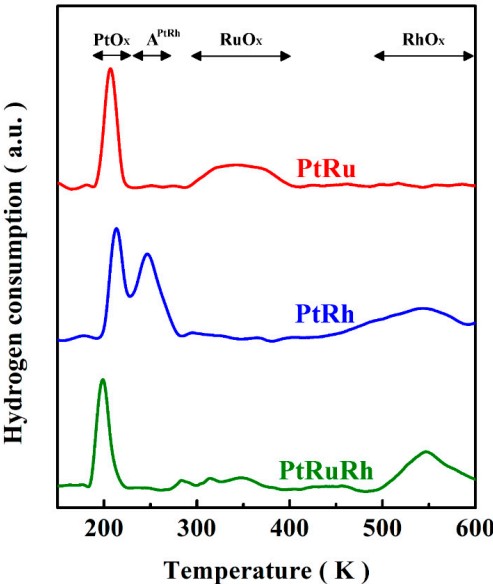

**Figure 3.** Temperature-programmed reduction (TPR) traces of PtRu, PtRh, and PtRuRh.

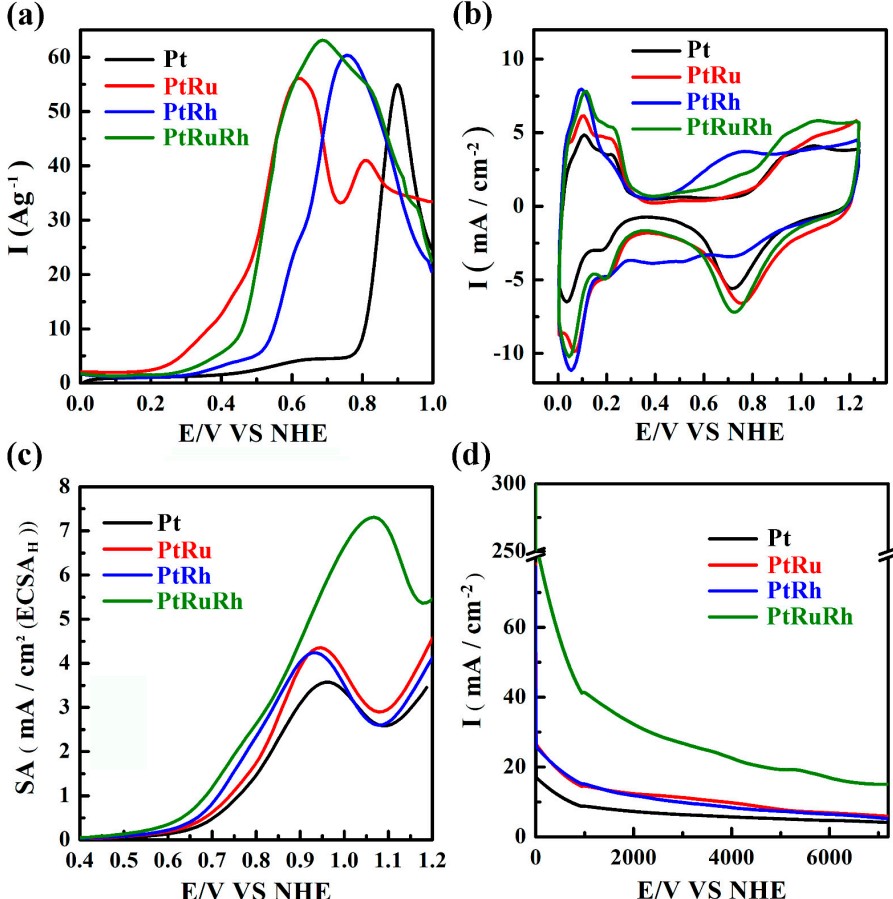

**Figure 4.** (**a**) CO-stripping voltammograms, (**b**) CV diagrams, (**c**) ethanol oxidation reaction (EOR) performance, and (**d**) CA curves of catalysts at a constant potential of 0.6 V vs. the normal hydrogen electrode (NHE) for Pt, PtRu, PtRh, and PtRuRh.

On the other hand, the CV diagrams of Pt, PtRu, PtRh, and PtRuRh catalysts are shown in Figure 4b, and the areas under the curves of the hydrogen desorption region used to calculate the

$ECSA_H$ for each catalyst are displayed in Table 1. The $ECSA_H$ of ternary catalysts are also larger than those of Pt and binary ones, suggesting better Pt utilization from the former. However, when comparing the calculated $ECSA_{CO}$ with $ECSA_H$, large differences are noted for the same catalysts. This phenomenon has been reported by Vidaković et al., who suggest that the uncertainty of the CO-stripping method is the type of the CO bonding on the surface, rather than the H-adsorption method giving the maximum number of surface reactive Pt sites [51]. As a result, the $ECSA_H$ is relatively accurate compared to $ECSA_{CO}$.

**Table 1.** EOR performances of Pt, PtRu, PtRh, and PtRuRh catalysts.

| Sample | $ECSA_H$ $(m^2/g_{(Pt)})$ | $I_{06}$ | $I_{max}$ | $SA_{06}$ | $SA_{max}$ | $I_f/I_b$ | $I_{06-7200}$ $(A/g_{(Pt)})$ |
|---|---|---|---|---|---|---|---|
| | | $(A/g_{(Pt)})$ | | $(mA/cm^2(ECSA_H))$ | | | |
| Pt | 30.8 | 6.6 | 178.8 | 0.15 | 3.57 | 0.83 | 4.1 |
| PtRu | 32.3 | 9.4 | 223.7 | 0.20 | 4.35 | 0.97 | 5.9 |
| PtRh | 33.0 | 10.9 | 214.3 | 0.24 | 4.24 | 1.06 | 5.2 |
| PtRuRh | 60.2 | 29.4 | 603.9 | 0.38 | 7.31 | 1.02 | 15.1 |

The EOR performances of Pt, PtRu, PtRh, and PtRuRh catalysts are summarized in Table 1, and their activities are shown in Figure 4c. All current densities are normalized by the $ECSA_H$ and the potentials are also adjusted to the normal hydrogen electrode (NHE). Besides, specific activity at 0.6 V ($SA_{06}$), maximum specific activity ($SA_{max}$), mass activity (MA) at 0.6 V ($I_{06}$) and the maximum MA ($I_{max}$) of catalysts are also compared in Table 1. It can be observed that the $SA_{06}$ of the PtRuRh catalyst is 0.38 mA/cm$^2$, which is 2.5, 1.9, and 1.6 times higher than that of Pt, PtRu, and PtRh, respectively and superior to PtRu and PtRh-based ternary catalysts as compared in Table S4. In terms of MA, the $I_{06}$ of PtRuRh catalyst is 29.4 A/g$_{(Pt)}$, which is 4.4, 3.1, and 2.7 times higher than that of Pt, PtRu, and PtRh, respectively. The rate determining step in the low potential region, such as 0.6 V is the dissociative adsorption of ethanol on surface Pt active sites [64–66]; therefore, high $SA_{06}$ and $I_{06}$ of PtRuRh also imply that the effective capability to dissociate ethanol on Pt, attributed to the high ECSA and low onset potential towards CO-stripping. On the other side, the binary and ternary catalysts all provide higher $SA_{max}$ than Pt alone; especially PtRuRh, which has the $SA_{max}$ of 7.31 mA/cm$^2$, owing to the presence of RuOx [67,68] and RhOx [69] to promote EOR through the bifunctional mechanism, as reported in Table S3 and Figure 4c. Besides EOR performance, the forward anodic peak current density (If) to the backward peak current density (Ib), If/Ib ratios of binary and ternary [70] are also promoted due to the addition of the second and third metals. As can be seen in Table 1, If/Ib ratio increases from 0.83 to 0.97–1.02, suggesting a better catalytic tolerance to carbonaceous species for the binary and ternary catalysts.

Moreover, in order to evaluate the stability of Pt, PtRu, PtRh, and PtRuRh catalysts, CA tests were carried out at 0.6 V in 0.5 M $H_2SO_4$ solution containing 1 M ethanol for 7200 s and the results are exhibited in Figure 4d, and the MAs at 0.6 V after 7200 s ($I_{06-7200}$) are summarized in Table 1. As shown in Figure 4d, catalysts decay to different extents during CA tests owing to the poisoning of Pt surface active sites. Among them, PtRuRh shows the highest current density of 15.1 A/g$_{(Pt)}$ after EOR for 7200 s, due to the bi-functional mechanism by the surface Pt/Ru/Rh, where Pt provides active sites for ethanol adsorption and dissociation, and Ru/Rh provide OCS for oxidating and removing the adsorbed intermediates formed during ethanol oxidation [65,71].

## 4. Conclusions

In this study, in order to prepare highly effective EOR catalysts, Pt binary and ternary catalysts such as PtRu, PtRh, and PtRuRh have been synthesized. The addition of Ru and/or Rh in Pt can effectively promote the anti-poisoning ability and help the desorptions of adsorbed intermediates, especially for PtRu. Moreover, EOR activities are enhanced significantly through Ru and/or Rh addition, especially ternary PtRuRh with the Pt/Ru/Ru ratio of 80/5/15, which both possess the highest $I_{06}$ and

$I_{06-7200}$. This can be attributed to the co-existence of $RuO_x$ and $RhO_x$ onto surface that not only helps the formations of $CO_2$ but also the removals of adsorbed intermediates, preventing Pt active sites from poisoning. Based on the above results, we can conclude that the addition of Ru and Rh into Pt catalysts can indeed enhance the activity and long-term performance of EOR. Therefore, carbon-supported PtRuRh NRs can be one of the most promising EOR catalysts.

**Supplementary Materials:** The following are available online at http://www.mdpi.com/2076-3417/10/11/3923/s1, Figure S1: XRD patterns, Table S1: SEM-EDS and XRD results, Table S2: XPS results, Table S3: CO-stripping results, and Table S4: comparison of EOR activity.

**Author Contributions:** Conceptualization, T.-H.H. and Y.-R.H.; methodology, T.-H.H.; validation, D.B., and S.L.; writing—original draft preparation, T.-H.H. and Y.-R.H.; writing—review and editing, T.-H.H. and K.-W.W. supervision, K.-W.W.; project administration, K.-W.W.; funding acquisition, K.-W.W. All authors have read and agreed to the published version of the manuscript.

**Funding:** This research was funded by Ministry of Science and Technology, R.O.C. (MOST 107-2628-E-008-003-MY3, MOST 108-3116-F-008-008, and MOST 108-3116-F-007-001. The authors like to thanks Prof. T.-Y. Chen for the funding support (Ministry of Science and Technology, Taiwan (MOST 109-3116-F-007-001 and MOST 106-2112-M-007-016-MY3)).

**Conflicts of Interest:** The authors declare no conflict of interest.

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
