# Peer review of "The Ethanol Oxidation Reaction Performance of Carbon-Supported PtRuRh Nanorods"

_applsci, doi:10.3390/app10113923_

Round 1
Reviewer 1 Report
The paper focuses on the study of carbon-supported Pt-based catalysts, including PtRu, PtRh, and PtRuRh nanorods for ethanol oxidation reaction.
The reported results are interesting and the paper in the overall is well written, with significant focus on catalyst characterization.
The reference list is adequate
The paper can be accepted for publication.
Few comments are reported in the following:
- Line 18, explicitate the acronym CA
- Line 51, please correct the word catalyst
- Line 78, Please explicitate the acronym MOR
- Lines 130-133, explicitate the purpose of Bragg’s law
- Line 134, please replace “eq 2.2” with “eq. (7)”
Author Response
Thanks for the valuable comments from the reviewer. We have made some revisions in accordance with the reviewers’ comments. We will describe in sequence in the response file and mark in the manuscript (in blue color).
- Line 18, explicitate the acronym CA
Response: CA is chronoamperometry (CA).
- Line 51, please correct the word catalyst
Response: The mistake has been corrected.
3 Line 78, Please explicitate the acronym MOR
Response: MOR is methanol oxidation reaction.
- Lines 130-133, explicitate the purpose of Bragg’s law
Response: In order to calculate d-spacing, Bragg's law was applied.
- Line 134, please replace “eq 2.2” with “eq. (7)”
Response : The mistake has been corrected.
Reviewer 2 Report
This paper reports on the electrooxidation reaction of ethanol in acidic media over the Pt based tri- and/or bi- metallic compounds. I understand the prepared ternary PtRuRh electrocatalyst has a superior electrocatalytic activity towards ethanol oxidation compared with Pt, PtRu and PtRh. Although, I think that the presented data and the text are well organized, there are several concerns. I recommend the publication in Applied Science after the addressing following.
The CO-stripping test is strange. I see that the CO-stripping voltammograms was carried out under the CO-saturate in H2SO4 electrolyte aqueous solution. Actually, CO dissolved in electrolyte must be removed by the bubbling, for example, of Ar for 15 – 20 mins. In the current case, CO dissolved in electrolyte was also electro-oxized on the working electrode, resulting that accurate ECSA(CO) and CO-tolerance ability cannot be predicted. Authors should read carefully the other reports which mention the method of CO-stripping test.
In Figure 4c, the shoulder peak could be slightly observed at ca. +0.75 V in the case of PtRh and PtRuRh. How dose author think this electrochemical phenomenon? The electrocatalysts that contain Rh differ the reaction mechanism for ethanol oxidation?
Authors insist main surface species of PtRuRh is Pt/Ru/Rh based on the TPR experiments. However, I think the surface structure is changed under electrochemical measurement condition.
The authors should compare catalytic activity with other literatures on the electrocatalytic activity of EtOH oxidation.
Minor points
- The unit (g^-2) in Fig. 4a should be changed to g^-1.
- Please add a more detail of method of EtOH electrooxidation (Fig. 4c) in experimental section (2.2).
- I think the specific activity is better in unit title in Fig. 4c.
- Authors should define MA, If, Ib, I06, I06-7200 and so on.
Author Response
Thanks for the valuable comments from the reviewer. We have made some revisions in accordance with the reviewers’ comments. We will describe in sequence in the response file and mark in the manuscript (in blue color).
- The CO-stripping test is strange. I see that the CO-stripping voltammograms was carried out under the CO-saturate in H2SO4 electrolyte aqueous solution. Actually, CO dissolved in electrolyte must be removed by the bubbling, for example, of Ar for 15 – 20 mins. In the current case, CO dissolved in electrolyte was also electro-oxized on the working electrode, resulting that accurate ECSA(CO) and CO-tolerance ability cannot be predicted. Authors should read carefully the other reports which mention the method of CO-stripping test.
Response:Thanks for the comment of the reviewer. The experimental procedure has been revised as suggested “CO-stripping was operated by purging CO gas in 0.5 M H2SO4 for 30 mins before the experiment; while operating, CO was kept purging at -0.1 V (vs. SCE) for 30 mins, then CO-stripping was measured between -0.3 and 0.76 V (vs. SCE) at the rate of 50 mV s-1 in N2 saturated 0.5 M H2SO4.”
- In Figure 4c, the shoulder peak could be slightly observed at ca. +0.75 V in the case of PtRh and PtRuRh. How dose author think this electrochemical phenomenon? The electrocatalysts that contain Rh differ the reaction mechanism for ethanol oxidation? Authors insist main surface species of PtRuRh is Pt/Ru/Rh based on the TPR experiments. However, I think the surface structure is changed under electrochemical measurement condition. The authors should compare catalytic activity with other literatures on the electrocatalytic activity of EtOH oxidation.
Response: The comparison of EOR activity between PtRuRh and other PtRu and PtRh based ternary catalysts taken from literature has been added in Table S4.
The following results and discussion have been revised.
“The EOR performances of Pt, PtRu, PtRh, and PtRuRh catalysts are summarized in Table 1, and their activities are shown in Figure 4 (c). All current densities are normalized by the ECSAH and the potentials are also adjusted to the normal hydrogen electrode (NHE). Besides, specific activity at 0.6 V (SA06), maximum specific activity (SAmax), mass activity (MA) at 0.6 V (I06) and maximum MA (Imax) of catalysts are also compared in Table 1. It can be observed that SA06 of PtRuRh catalyst is 0.38 mA/cm2, which is 2.5, 1.9, and 1.6 times higher than that of Pt, PtRu, and PtRh, respectively. In terms of MA, the I06 of PtRuRh catalyst is 29.4 A/g(Pt), which is 4.4, 3.1, and 2.7 times higher than that of Pt, PtRu, and PtRh, respectively.”
Minor comments
1.The unit (g^-2) in Fig. 4a should be changed to g^-1.
Response: The mistake has been correct in Figure 4A.
2. Please add a more detail of method of EtOH electrooxidation (Fig. 4c) in experimental section (2.2).
Response: For the EOR activity, the CVs were swept between -0.24 to 1.0 V (vs. SCE) with a scanning rate of 2 mVs-1 in 0.5 M H2SO4 and 1 M C2H5OH saturated with N2.
3. I think the specific activity is better in unit title in Fig. 4c.
Response: Figure 4C has been revised as suggested.
4. Authors should define MA, If, Ib, I06, I06-7200 and so on.
Response: MA is mass activity
If is the forward anodic peak current density and Ib is the backward peak current density.
I06 is the mass activity at 0.6 V.
I06-7200 is the mass activity at 0.6 V after EOR for 7200 s.
Round 2
Reviewer 2 Report
The CO stripping voltammogram here is still not that convincing. If CO stripping test was carried out in N2 saturated H2SO4 solution, why doublet peaks could be observed at conventional Pt electrocatalyst? Please explain it before it gets acceptance.
Author Response
Thanks for the comments of the reviewer.
The reviewer is absolutely correct in this point.
We are very sorry for the mistake. The CO-stripping results of Pt have been revised in Figure 4.
We hope that the revisions and explanation are satisfactory and the paper can be reconsidered for publication. The reviewer has been very thorough and we appreciate that.
